# Supplementation of Probiotic *Butyricicoccus pullicaecorum* Mediates Anticancer Effect on Bladder Urothelial Cells by Regulating Butyrate-Responsive Molecular Signatures

**DOI:** 10.3390/diagnostics11122270

**Published:** 2021-12-04

**Authors:** Yen-Chieh Wang, Wei-Chi Ku, Chih-Yi Liu, Yu-Che Cheng, Chih-Cheng Chien, Kang-Wei Chang, Chi-Jung Huang

**Affiliations:** 1Department of Urology, Cathay General Hospital, Taipei 106438, Taiwan; litiger.wang@msa.hinet.net; 2School of Medicine, College of Medicine, Fu Jen Catholic University, New Taipei 242062, Taiwan; 089052@mail.fju.edu.tw (W.-C.K.); cyl1124@gmail.com (C.-Y.L.); yccheng@cgh.org.tw (Y.-C.C.); chienc@cgh.org.tw (C.-C.C.); 3Department of Pathology, Sijhih Cathay General Hospital, New Taipei 221037, Taiwan; 4Department of Medical Research, Cathay General Hospital, Taipei 106438, Taiwan; 5Department of Biomedical Sciences and Engineering, National Central University, Taoyuan 320317, Taiwan; 6Department of Anesthesiology, Cathay General Hospital, Taipei 106438, Taiwan; 7Taipei Neuroscience Institute, Taipei Medical University, Taipei 110301, Taiwan; kwchang@tmu.edu.tw; 8Laboratory Animal Center, Taipei Medical University, Taipei 110301, Taiwan; 9Department of Biochemistry, National Defense Medical Center, Taipei 114201, Taiwan

**Keywords:** urothelial bladder cancer, butyrate, *Butyricicoccus pullicaecorum*, apoptosis, bladder cancer-associated protein

## Abstract

In bladder cancer, urothelial carcinoma is the most common histologic subtype, accounting for more than 90% of cases. Pathogenic effects due to the dysbiosis of gut microbiota are localized not only in the colon, but also in regulating bladder cancer distally. Butyrate, a short-chain fatty acid produced by gut microbial metabolism, is mainly studied in colon diseases. Therefore, the resolution of the anti-cancer effects of butyrate-producing microbes on bladder urothelial cells and knowledge of the butyrate-responsive molecules must have clinical significance. Here, we demonstrate a correlation between urothelial cancer of the bladder and *Butyricicoccus pullicaecorum*. This butyrate-producing microbe or their metabolite, butyrate, mediated anti-cancer effects on bladder urothelial cells by regulating cell cycle, cell growth, apoptosis, and gene expression. For example, a tumor suppressor against urothelial cancer of the bladder, bladder cancer-associated protein, was induced in butyrate-treated HT1376 cells, a human urinary bladder cancer cell line. In conclusion, urothelial cancer of the bladder is a significant health problem. To improve the health of bladder urothelial cells, supplementation of *B. pullicaecorum* may be necessary and can further regulate butyrate-responsive molecular signatures.

## 1. Introduction

The burden of urologic cancer is increasing globally because of population aging [1]. In addition to prostate cancer, other urologic cancers such as bladder cancer and kidney cancer remain common malignancies worldwide [2]. Among these urinary system tumors, urothelial carcinoma displays distinct histomorphological phenotypes [3,4] and is likely to arise from different uroprogenitor cells [5]. Genomic characterization of bladder cancer may provide new insights into understanding the nature of this complex disease, leading to the development of effective therapies [3,5].

The microbiota isolated from the urine of patients with bladder cancer have been associated with bladder tumors [6,7,8]. Some of these urine-derived bacteria may potentially become biomarkers or therapeutic targets for bladder cancer [9,10]. Most microbiota are located within the colonic tract, where they play a clinically significant role in human health and the development of disease. However, the effects of gut microbiota are also found on other organs, where they exert pathogenic effects as a result of dysbiosis [11,12,13,14]. Dysbiosis of bacteria in the colon is thought to be involved in regulating bladder cancer [15,16]. Some ongoing clinical trials have focused on the signatures of gut microbiota as markers of the efficacy of immune-checkpoint immunotherapy in bladder cancers [17]. Short-chain fatty acids (SCFAs), a group of metabolites from gut microbiota, have been known to play a crucial role in improving human health for many years [18]. For example, one SCFA, butyrate, synthesized from dietary carbohydrates by bacterial fermentation in the colon [15,19], may be a potent proliferation inhibitor of urothelial cancer of the bladder [15,20]. In this context, butyrate-induced growth inhibition is potentially clinically significant [20]. In addition, butyrate may influence the expression of tumor suppressors involved in chemotherapy treatment, affecting the pathological outcome of some bladder cancers [21]. Therefore, a better understanding of the role of butyrate-producing microbes and the correlation between butyrate and tumor suppressors in the development and progression of bladder cancer would be helpful for exploring new therapeutic options.

In fact, a safe butyrate-producing microbe, *Butyricicoccus pullicaecorum*, has been shown in clinical trials to possibly reduce cancer progression [22,23]. In the present research, we aimed to demonstrate the anticancer effect of *B. pullicaecorum* on the induction of antitumor molecular events by determining the molecular effects of butyrate on urothelial cancer of the bladder and evaluating cell cycle regulation and apoptosis in the cells of bladder urothelial cancer in the presence of butyrate. Our results suggest that the major metabolite of butyrate-producing microbes in the colon (butyrate) is an important factor in impairing the development of bladder cancer. Supplementation of *B. pullicaecorum* may provide opportunities for therapeutic intervention in different bladder cancers through the secretion of the specific SCFA, butyrate.

## 2. Materials and Methods

### 2.1. Mouse with B. pullicaecorum Administration

C3H/He mice (male) aged 4–6 weeks were provided by the National Laboratory Animal Center, NARLabs, Taiwan. All protocols were approved by the Institutional Animal Care and Use Committees at Cathay General Hospital (Taipei, Taiwan). All efforts were made to minimize the number of animals and their suffering. Three to five mice were housed per plastic cage under pathogen-free conditions (humidity: 50% ± 10%; light: 12/12 h light/dark cycle; temperature: 23 °C ± 2 °C) in an individually ventilated cage rack system (Tecniplast, Varese, Italy). Mice were randomly divided into two groups for testing *B*. *pullicaecorum*: (1) control subgroup (*n* = 4) and (2) subgroup with *B*. *pullicaecorum* (1 × 10^7^ CFU in 100 μL) administration through anal injection (*n* = 4), as illustrated in Appendix A. *B*. *pullicaecorum* (BCRC-81109) was purchased from the Bioresource Collection and Research Center (Hsinchu, Taiwan) and grown in BCRC medium 967 for 3 days in anaerobic conditions at 37 °C. The efficacy of *B. pullicaecorum* administration was acquired by specific quantitative real-time PCR (Appendix A).

### 2.2. Urinary Bladder Cells and Sodium Butyrate Treatment

Human urinary bladder cancer cell line HT1376 (ATCC CRL-1472) was purchased from American Type Culture Collection (ATCC; Manassas, VA, USA). This cell line was expanded in a complete medium (Eagle’s Minimum Essential Medium with 10% fetal bovine serum) under 95% air atmosphere (with CO_2_) in a 37 °C humidified incubator. To evaluate the effects of butyrate on HT1376 cells, the 50% inhibitory concentration (IC_50_) of sodium butyrate (NaB) was used to treat cells for 72 h.

### 2.3. Immunohistochemical Staining

For immunohistochemical staining to detect various proteins, a VECTASTAIN Elite ABC kit (Cat No. PK-6101; Vector Laboratories, Burlingame, CA, USA) was used according to the manufacturer’s instructions. Briefly, after deparaffinization, sections were rehydrated sequentially with 100%, 90%, and 70% ethanol. These rehydrated slides were immersed in a citrate buffer (10 mM, pH 6.0), boiled (95–99 °C) for 20 min and then cooled to 25 °C for 20 min. To inactivate endogenous peroxidases and block potential nonspecific binding sites, tissue sections were incubated for 30 min in a 3% H_2_O_2_ methanolic solution and then blocked for a further 30 min with a blocking serum solution (Vector Laboratories). Target protein was, respectively immune-detected with anti-GPR43 antibody (1:50 in blocking solution; Cat No. BS-13536R; Bioss, Boston, MA, USA), anti-GPR109B antibody (1:500 in blocking solution; Cat No. ABP-56889; Abbkine, Wuhan, China), anti-BLCAP antibody (1:10 in blocking solution; Cat No. PA5-38639; Thermo Fisher Scientific, Waltham, MA, USA), and anti-FABP4 antibody (1:25 in blocking solution; Cat No. 15872-1-AP; Proteintech, Rosemont, IL, USA) for 16 h at 4 °C. After several washes in Tris-buffered saline, tissue sections were incubated with biotinylated goat anti-rabbit IgG antibody (1:200 in blocking solution; cat no. BA-1000; Vector Laboratories) for 0.5 h at 25 °C. Finally, all target proteins were visualized using the DAB Substrate Kit (Cat No. SK-4100; Vector Laboratories) as the substrate. Hematoxylin was used to stain the nucleus, and the results were analyzed by pathologists.

### 2.4. Change in Growth of HT1376 Cells following NaB Treatment

The growth of HT1376 cells was evaluated using an MTT assay (Cat. No. M5655; Merck KGaA). First, HT1376 cells were cultured at 1 × 10^4^ cells/well in 96-well flat-bottom plates for 24 h and the IC_50_ of NaB was determined by treating cells at different concentrations (0.05, 0.5, 5, 50, 100, and 200 mM) for 72 h. To evaluate the efficiency of adjuvant therapy, HT1376 cells (5 × 10^3^ cells/well) were treated with a combined medium (the complete medium for cell culture and a certain percentage of the conditioned medium for *B*. *pullicaecorum* cultivation) for 3 or 6 days, and cell viability was calculated using an MTT assay. Briefly, the cells were treated with 10 μL of the MTT reagent and incubated in the dark for 4 h. Dimethyl sulfoxide (100 μL) was subsequently added to dissolve the purple precipitate formed by the viable cells. The absorbance of each well at 540 nm was read by a Synergy HT Multi-Mode microplate reader (BioTek Instruments). A 0.22 μm membrane filter (Cat No. GSWP04700; Merck KGaA) was used for sterilizing filtration before use. Data were obtained from three independent experiments.

For the bromodeoxyuridine (BrdU) incorporation experiment, HT1376 cells were treated with the predetermined IC_50_ of NaB and cultured for 72 h. Cells were then processed for BrdU incorporation using a BrdU cell proliferation assay Kit (Cat. No. K306-200; Biovision, Milpitas, CA, USA) according to the manufacturer’s protocol. Plates were read immediately after the addition of stop solution at 450 nm in triplicate in a Synergy HT Multi-Mode microplate reader (BioTek Instruments).

### 2.5. Quantitative PCR

Total RNA of HT1376 cells with the predetermined IC_50_ of NaB for 72 h was extracted using RNAzol RT (Molecular Research Center) and converted to cDNA with a High-Capacity cDNA Reverse Transcription Kit in the presence of oligo(dT) primers (Thermo Fisher Scientific) according to the manufacturer’s instructions. To quantify the expression of GPR43 and GPR109B, the reaction mixture containing the cDNA sample, QuantiTect SYBR-Green PCR Master mix (Qiagen GmbH), QuantiTect Primer assay (Cat No. Hs_FFAR2_1_SG for GPR43 and Cat No. Hs_HCAR3_1_SG for GPR109B; Qiagen GmbH), and RNase-free water was amplified using the following cycling program: 10 min at 95 °C followed by 40 cycles at 95 °C for 15 s and at 60 °C for 1 min. TaqMan Gene Expression Assays were applied by commercially available primer sets (FasL, Cat No. Hs00181225_m1; Thermo Fisher Scientific) or universal probe sets (FABP4, BLCAP, and CDK1; primer sequences and universal probe number in Table 1; Roche Diagnostics GmbH). LightCycler TaqMan Master (Roche Diagnostics GmbH) and RNase-free water was amplified by a program (10 min at 95 °C, proceeding with 60 cycles at 95 °C for 10 s and at 60 °C for 20 s). All mRNA levels were adjusted relative to the level of glyceraldehyde-3-phosphate dehydrogenase (Table 1) and all quantitative PCR reactions were run in a LightCycler 96 (Roche Diagnostics GmbH) and the data were analyzed using the 2−ΔΔCq method [24].

### 2.6. Analyses of Cell Cycle and Apoptosis through Image Cytometry

The cell cycle and apoptosis analyses of HT1376 cells were performed using a fluorescence image cytometer (NucleoCounter NC-3000; ChemoMetec A/S, Denmark) [25,26]. HT1376 cells were seeded in a six-well dish at a density of 3.6 × 10^5^ cells per well and treated with the predetermined IC_50_ of NaB for 72 h after a 16 h pre-incubation period.

For cell cycle analysis, cells were harvested by trypsinization, suspended in 0.5 mL of phosphate-buffered saline (PBS) and fixed with 4.5 mL of 70% cold ethanol for at least 2 h. Subsequently, the ethanol was removed and the cells were resuspended in PBS. Cell pellets were harvested by centrifugation at 500× *g* for 5 min at 4 °C and incubated with 0.5 mL DAPI solution (0.1% Triton X-100 and 4′,6-diamidino-2-phenylindole) for 5 min at 37 °C. The stained cells were loaded into an NC-Slide A8 (ChemoMetec) and evaluated using a Fixed Cell Cycle-DAPI/DNA fragmentation assay protocol in the NucleoCounter NC-3000 image cytometer (ChemoMetec) [27]. The acquired DNA content histograms were used to distinguish cells at different phases (G1, S, and G2/M) of the cell cycle with NucleoView NC-3000 software (version 2.1.25.12; ChemoMetec).

For the apoptosis analysis, cells were harvested by trypsinization and suspended in 0.1 mL of annexin V binding buffer with 2 μL of FITC-labeled annexin V and 2 μL of Hoechst33342 (500 μg/mL). The cells were then incubated for 15 min at 37 °C and subsequently centrifuged at 400× *g* for 5 min. Following the removal of the supernatant, the cell pellets were resuspended in 300 μL of annexin V binding buffer and centrifuged twice under the conditions described above. The cell pellets were resuspended in 100 μL of annexin V binding buffer, and 2 μL of propidium iodide (500 μg/mL) was added. Eventually, the prepared samples were loaded immediately into NC-Slide A2 (ChemoMetec) and analyzed using the annexin V assay protocol in the NucleoCounter NC-3000 image cytometer (ChemoMetec) [28]. The obtained scatterplots were used to demarcate the percentage of healthy cells and early and late apoptotic cells, respectively.

Western blots to detect FasL protein from HT1376 cells without or with 2.4 mM NaB treatment for 72 h were performed followngi standard procedure and the antibodies specific to target proteins were based on our previous report with minor modifications: anti-FasL, 1:20, #ab15285 (Abcam, Cambridge, UK); anti-glyceraldehydes-3-phosphate dehydrogenase, 1:4000, #AM4300 (Thermo Fisher Scientific) [29]. The secondary antibody, either anti-rabbit or anti-mouse, which was conjugated with alkaline phosphatase, was then used. Blots were finally developed using VECTASTAIN ABC-AmP DuoLuX chemiluminescent/fluorescent substrate kits for alkaline phosphatase (SK-6005; Vector Laboratories, Burlingame, CA, USA) according to the manufacturer’s instructions. The images of immunoblots were captured on a FluorChem FC2 system (Cell Biosciences, Santa Clara, CA, USA).

### 2.7. Statistical Analysis

Student’s *t*-test was used to determine the mean differences between treatment and control, and values of *p* < 0.05 were considered statistically significant. Data are presented as mean ± SEM.

## 3. Results

### 3.1. In Vivo Evaluation of SCFA-Related Gene Expression in Mouse Bladder after B. pullicaecorum Administration

Reduction of intestinal microbial metabolite butyrate due to gut microbiota imbalance was believed to be related to colorectal cancer (CRC) development [30]. We also reported that gut microbe *B*. *pullicaecorum* could regulate the SCFA transporter and receptor to reduce CRC progression [23]. Thus, we determined the protein level of different SCFA-related genes in mouse bladder urothelial cells after *B*. *pullicaecorum* administration, which increased the level of *B. pullicaecorum* in stools (Appendix A). From immunohistochemistry, the predominant dense-intensity staining was seen in transitional epithelium and some positive signals in the lamina propria of GPR43 (Figure 1A), GPR109B (Figure 1B), and FABP4 (Figure 1C). In addition, as illustrated in Figure 1, GPR43, GPR109B, and FABP4 staining existed broadly and evenly in the transitional epithelium of the urinary bladder of mice with this *B*. *pullicaecorum* supplementation. By contrast, weak signals of these SCFA-related proteins were locally detected from the bladder urothelial cells of mice when mice were not administered *B*. *pullicaecorum*.

### 3.2. Upregulation of Butyrate-Responsive Genes in Urothelial Cancer Cells of the Bladder after NaB Treatment

We then used bladder urothelial cancer cell line HT1376 to elaborate the effects of NaB on changes of butyrate-responsive molecular signatures. We initially performed the MTT assay to calculate the total cell numbers following the dose dependence and acquired the IC_50_ after 72 h NaB exposure. An IC_50_ value of about 2.4 mM against HT1376 cells was observed (Figure 2A). The mRNA levels of two G protein-coupled receptors, i.e., GPR43 (11.4-fold) and GPR109B (2.8-fold), were significantly upregulated in HT1376 cells after NaB treatment at IC50 (Figure 2B: GPR43, *p* < 0.01; Figure 2C: GPR109B, *p* < 0.05) compared with cells without NaB treatment. Similarly, compared with HT1376 cells not treated with NaB, the NaB-treated HT1376 cells exhibited a higher mRNA level of the fatty acid binding protein, FABP4 (107.7-fold; Figure 2D, *p* < 0.01).

### 3.3. Induction of Apoptosis in Bladder Urothelial Cells under a Butyrate-Enriched Microenvironment

Molecular and cellular functions of butyrate were further studied with in vitro NaB-treated HT1376 cells and in vivo bladder urothelial cells of *B*. *pullicaecorum*-supplemented mice. The expression level of the bladder cancer-associated protein (BLCAP), which is an apoptosis-related tumor suppressor identified from bladder cancer [31,32], was significantly higher (29.2-fold; *p* < 0.05) in NaB-treated HT1376 cells than that in cells without NaB treatment (Figure 3A). Moreover, the immunoreactivity of BLCAP was observed in the layer of transitional epithelium of mouse urinary bladder regardless of whether mice had been treated with *B*. *pullicaecorum*. Nonetheless, mice with NaB treatment had stronger immunoreactivity of BLCAP in the transitional epithelium and lamina propria than did control mice, as illustrated in Figure 3B. The improved apoptosis induced by a butyrate-enriched microenvironment could also be found in vitro from the annexin V/PI protocol [33]. As shown in Figure 3C, the percentage of late apoptotic HT1376 cells (annexin V- and PI-positive cells) increased from 10.7% to 22.8% after a 72 h NaB incubation period. In addition, we determined the level of Fas Ligand (FasL), one of the molecules involved in the regulation of cell death [34]. We quantified a relatively higher expression of FasL (2.9-fold, *p* < 0.05) (Figure 3D) and FasL protein level (Figure 3E) from the NaB-treated HT1376 cells compared with non-treated control cells.

### 3.4. NaB-Induced Inhibition of Cell Proliferation or Growth of Bladder Urothelial Cancer Cells

Cell growth is an essential requirement for cell cycle progression, but cell cycle arrest may inhibit growth [35]. In the present study, we found that NaB could significantly reduce the expression of CDK1 (0.5-fold) in HT1376 cells (Figure 4A) and arrest cells at the G2/M phase (31%), relative to the cells without NaB treatment (9%) (Figure 4B). In addition, we used the BrdU incorporation assay to evaluate the proliferation of attached HT1376 cells under NaB treatment. NaB treatment not only significantly reduced cell growth (Figure 5A) but also blunted the BrdU incorporation rate in HT1376 cells (Figure 5B). Similarly, cell growth was decelerated when the cell culture medium contained the conditioned medium of *B. pullicaecorum* cultivation. At Day 6 in Figure 5C, the cell growth slowed down from 2.0- to 1.2-fold (*p* < 0.01) when the medium contained only 5% conditioned medium, whereas cell growth was significantly reduced to 0.8-fold when it contained 10% conditioned medium (*p* < 0.001).

## 4. Discussion

The causes of bladder cancer are becoming increasingly complex and diverse. Urothelial carcinoma is the most common histologic subtype of bladder cancer and accounts for more than 90% of cases [36]. Epidemiological studies have identified that tobacco, hair dye, and occupational chemical exposure are believed to be responsible for urothelial carcinoma [37,38,39]. In addition to carcinoma, ketamine-associated ulcerative cystitis is another emerging bladder disease in some countries [40,41,42]. Therefore, it is of clinical significance whether there are substances that can prevent bladder cells from being threatened by bladder diseases. Here, we found that butyrate could modulate the growth of bladder urothelial cells and promote apoptosis. A supplementation of butyrate-producing *B*. *pullicaecorum* might improve the health of the urothelial bladder cells.

It is well-known that changes in the gut microbiota can mediate or modify the effects of environmental factors on the risk of CRC [43]. First, gut microbiota have been shown to play a role in promoting or developing CRC initiation [44,45]. For example, Fusobacterium nucleatum may affect carcinogenesis and extend to facilitating resistance to chemotherapy [46]. Second, there are potential benefits to using probiotics and prebiotics to modulate the host inflammatory response, as well as its application in CRC prevention and treatment [47,48]. For example, *B*. *pullicaecorum* is able to reduce CRC progression [23]. Butyrate is one of the SCFAs that are synthesized from dietary carbohydrates by bacterial fermentation in the colon [19]. Most SCFAs can be absorbed from the colonic lumen and used by colonic epithelial cells, while the remainder are transferred to circulation in different percentages [49,50,51]. This implies that butyrate in circulation can directly or indirectly function in various peripheral tissues [52] and this is interesting in terms of understanding the effects of butyrate on tissues other than the colon. Our present results are consistent with these reports. We showed that enough *B*. *pullicaecorum* in the colon affects the transitional epithelium of the urinary bladder to overexpress two receptors (GPR43 and GPR109B) and one transporter (FABP4).

With properties of anti-inflammation and being a histone deacetylase inhibitor, butyrate enhances barrier function and mucosal immunity in the colon through upregulation of the expression of SCFA receptors and transporters [23,53,54,55]. In this study, the increased expression of GPR43, GPR109B, and FABP4 was even detected in the transitional epithelium of the urinary bladder of mice treated with *B*. *pullicaecorum*. These molecular changes may contribute to the promotion and achievement of optimal bladder health. GPR43 and GPR109B are two SCFA-sensing G protein-coupled receptors, while GPR43 is a fatty acid receptor and GPR109B is a hydroxycarboxylic acid receptor [56]. Briefly, GPR43 is known to be activated by SCFAs and may improve anti-inflammatory activity in various tissues [57,58,59]. Here, we further showed that the transitional epithelium of urinary bladder of mice treated with *B*. *pullicaecorum* exhibited increased GPR43 synthesis, which was confirmed by the result of increased gene expression of GPR43 in NaB-treated HT1376 cells. Moreover, these SCFA receptors or transporters were also found to be upregulated in 5637 cells, which were also the bladder urothelial cancer cells (Appendix A). This implies that butyrate may increase the expression level of GPR43, GPR109B, and FABP4 in urothelial bladder cells. Butyrate also changed the ability of urothelial bladder cells to respond to the microenvironment.

Not only did we find an increase in GPR43 due to butyrate but perhaps more significantly we also found that the expression of GPR109B and FABP4 showed a similar expression pattern. The molecular effect of butyrate on GPR109B and FABP4 expression is different in different tissues. For example, GPR109B in gut epithelial cells has no expression difference between cancer and normal colon cells [60]. Nevertheless, breast cancer cells have higher GPR109B levels than do non-tumor control cells [61]. However, to our knowledge, this is the first study to determine the expression level of GPR109B in urothelial bladder cells due to butyrate treatment or *B*. *pullicaecorum* administration. GPR109B has been specifically identified as a receptor for the inhibition of lipolysis [62,63,64,65]. As reviewed by Zaidi et al., fatty acids acquired by lipolysis will nourish cancer cells [66]. Additionally, it has been found that tumor load positively correlates with intracellular lipolysis [67,68]. This increased lipolysis may attenuate the therapeutic advantage of human cancers [66,69]. Thus, Li et al. suggested that avoiding excessive lipolysis can be a new therapeutic strategy for cancer treatment [69]. Our data reveal that NaB treatment or *B*. *pullicaecorum* administration may increase GPR109B synthesis to protect bladder urothelial cells.

It has been confirmed in many studies that FABP4 is functionally responsible for aggressive patterns of colon cancer [70], breast cancer [71], ovarian cancer [72], prostate cancer [73], and non-small-cell lung cancer [74]. On the other hand, as reported by Zhong et al., the overexpression of FABP4 is able to repress tumor growth and invasion in hepatocellular carcinoma [75]. This result is consistent with the report of Chiu et al., who suggest that low-grade urothelial bladder cancer has higher FABP4 expression than high-grade cancers [76]. We found that the supplementation of *B*. *pullicaecorum* or butyrate treatment increased FABP4 synthesis in urothelial bladder cells, as reported by Mathis et al. [77]. Induction of FABP4 expression may prevent tumor progression. At the same time, we observed increased BLCAP in urothelial bladder cells, not only in the HT1376 cells but also in the 5637 cells (Appendix A). BLCAP exhibits a tumor suppressor function in various tumors, including urothelial cancer of the bladder [31] and can reduce cell growth by stimulating apoptosis [32]. The increase in BLCAP by *B*. *pullicaecorum* administration and butyrate treatment may reflect cell cycle arrest, FasL expression, and growth inhibition in NaB-treated bladder urothelial cancer cells.

Although we demonstrated that *B*. *pullicaecorum* supplementation changes the gene expression profile associated with the butyrate-regulated anticancer effect on bladder urothelial cancer, there are limitations to this study that should be noted. First, the possible effect of whole metabolites derived from *B*. *pullicaecorum* on bladder cancer was not fully evaluated, especially in combination with other anticancer drugs. For example, Maruyama et al. also showed that butyrate administration strengthens the anti-cancer effect of cisplatin, which is a standard chemotherapy for advanced bladder cancer [78]. Second, the potential interaction of the probiotic *B*. *pullicaecorum* with other gut microbiota and their effect on the efficacy of anti-cancer therapy was not addressed in this study. For instance, Aso and Akazan reported that oral administration of probiotics decreases superficial bladder cancer recurrence [18]. Manipulating gut microbiota, is also emerging as a promising adjuvant treatment to enhance the therapeutic effect of cancer immunotherapy [79,80]. Taken together, the therapeutic effect of *B*. *pullicaecorum* supplementation on the efficacy of anticancer drugs in bladder urothelial cancers warrants future studies.

## 5. Conclusions

SCFAs, such as butyrate, that are produced by gut microbial metabolism is commomly studied in colon diseases. In this study, we demonstrated that butyrate derived from *B*. *pullicaecorum* supplementation changes gene expression in bladder urothelial cancer cells in vitro. The butyrate-responsive molecular signatures correlate with the cellular growth of bladder urothelial cancer cells. Our results suggest that butyrate-producing probiotics, such as *B*. *pullicaecorum*, mediate the anticancer effect on bladder urothelial cells by increasing the anti-inflammation activity and anti-oncogenic potential of bladder urothelial cells.

## Figures and Tables

**Figure 1 diagnostics-11-02270-f001:**
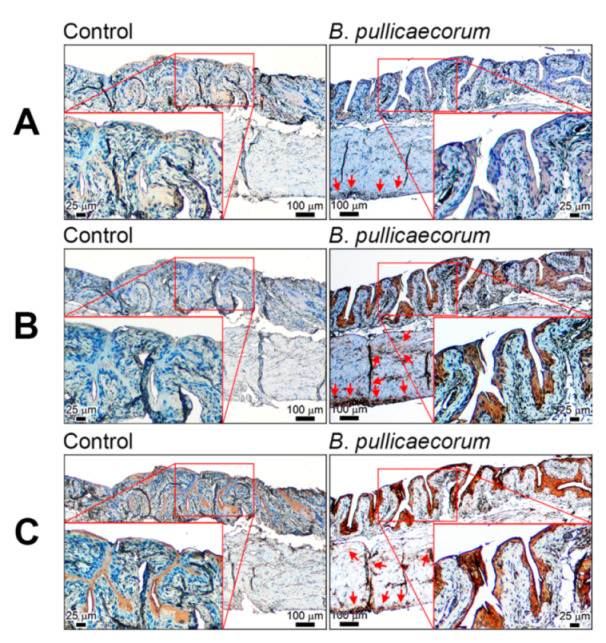
Representative images of SCFA-related gene expression in mouse bladder after *B. pullicaecorum* administration. (**A**) GPR43 immunohistochemistry staining. (**B**) GPR109B immunohistochemistry staining. (**C**) FABP4 immunohistochemistry staining. Control mice (*n* = 4), without *B. pullicaecorum* administration; *B. pullicaecorum* mice (*n* = 4), with 1 × 10^7^ CFU of *B. pullicaecorum* administration through anal injection. SCFA, short chain fatty acid.

**Figure 2 diagnostics-11-02270-f002:**
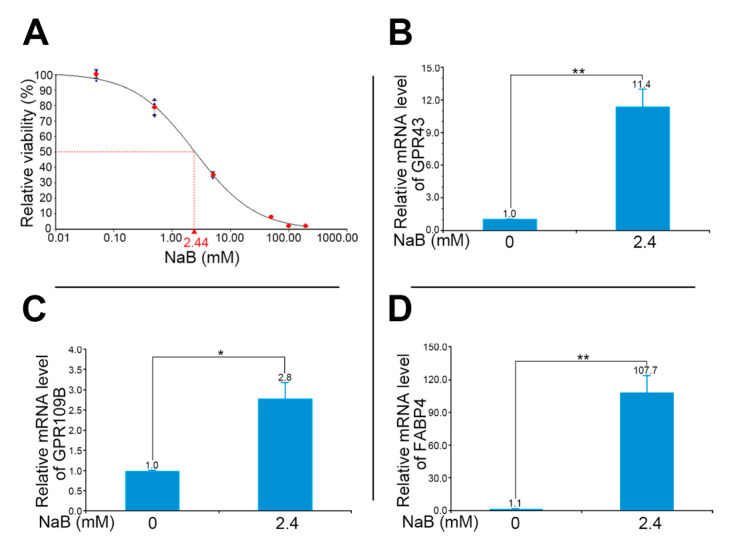
Up-regulation of SCFA-related genes in bladder urothelial cancer cells after NaB treatment. (**A**) The IC_50_ value of about 2.4 mM NaB against HT1376 cells. (**B**–**D**) Relative mRNA levels of GPR43, GPR109B, and FABP4 in HT1376 cells. All mRNA levels were adjusted relative to the level of glyceraldehyde-3-phosphate dehydrogenase. Data are the mean ± SEM of at least two independent experiments. SCFA, short chain fatty acid; NaB, sodium butyrate. * *p* < 0.05 and ** *p* < 0.01.

**Figure 3 diagnostics-11-02270-f003:**
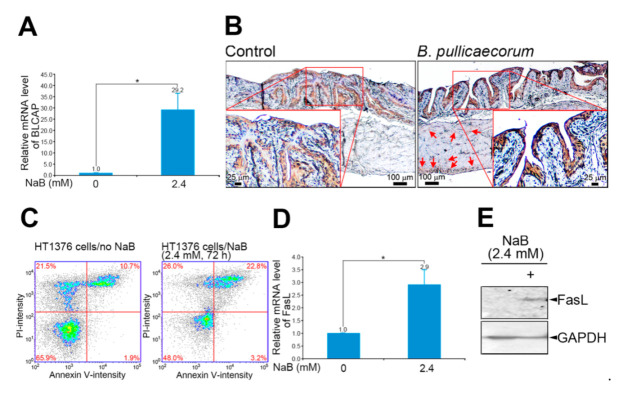
BLCAP and FasL levels in bladder urothelial cells after NaB treatment. (**A**) Relative mRNA level of BLCAP in HT1376 cells. The mRNA level of BLCAP was adjusted relative to the level of glyceraldehyde-3-phosphate dehydrogenase. (**B**) Representative images of BLCAP immunohistochemistry staining. Control mice (*n* = 4), without *B. pullicaecorum* administration; *B. pullicaecorum* mice (*n* = 4), with 1 × 10^7^
*B. pullicaecorum* administration through anal injection. (**C**) Apoptotic rate of HT1376 cells after NaB treatment. (**D**) Relative mRNA level of FasL in HT1376 cells. The mRNA level of CDK1 was adjusted relative to the level of glyceraldehyde-3-phosphate dehydrogenase. Data are the mean ± SEM of at least two independent experiments. (**E**) The protein expression of FasL in HT1376 cells. Data are the mean ± SEM of at least two independent experiments. NaB, sodium butyrate. * *p* < 0.05.

**Figure 4 diagnostics-11-02270-f004:**
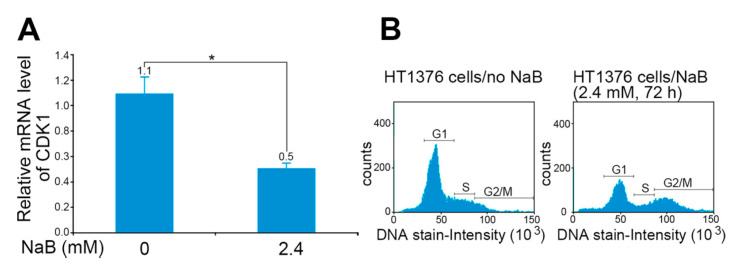
Regulation of cell cycles in bladder urothelial cancer cells by NaB. (**A**) Relative mRNA level of CDK1 in HT1376 cells. (**B**) Cell cycle distribution of HT1376 cells after NaB treatment. Data are the mean ± SEM of at least two independent experiments. NaB, sodium butyrate. * *p* < 0.05.

**Figure 5 diagnostics-11-02270-f005:**
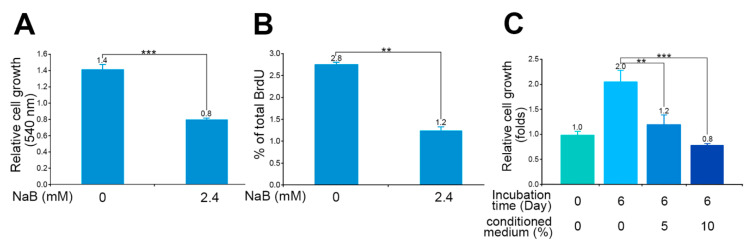
Inhibition of cell growth of bladder urothelial cancer cells by NaB. (**A**) Relative growth of HT1376 cells after NaB treatment. (**B**) Change of BrdU incorporation rate in HT1376 cells after NaB treatment. (**C**) Relative growth of HT1376 cells after conditioned medium treatment. Data are the mean ± SEM of at least two independent experiments. NaB, sodium butyrate. ** *p* < 0.01 and *** *p* < 0.001.

**Table 1 diagnostics-11-02270-t001:** Primer sequences and probe numbers for real-time PCR.

Gene Name	Accession Number	Sequence (From 5′ to 3′)	UPL Number
FABP4	NM_001442	F: CCACCATAAAGAGAAAACGAGAG	#31
		R: GTGGAAGTGACGCCTTTCAT	
BLCAP	NM_006698	F: CGCCATGGTTCCAAGAAT	#17
		R: CGCTTTCTTCAACCCTCACT	
CDK1	NM_001786	F: TGGATCTGAAGAAATACTTGGATTCTA	#79
		R: CAATCCCCTGTAGGATTTGG	
GAPDH	NM_002046	F: CTCTGCTCCTCCTGTTCGAC	#60
		R: ACGACCAAATCCGTTGACTC	

ABP4, fatty acid binding protein 4; BLCAP, bladder cancer associated protein; CDK1, cyclin dependent kinase 1; GAPDH, glyceraldehyde-3-phosphate dehydrogenase; F, forward primer; R, reverse primer; UPL, Universal ProbeLibrary from Roche Diagnostics GmbH (Germany).

## Data Availability

The raw data supporting the conclusions of this article will be made available by the authors without undue reservation.

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
