# Peer review of "Supplementation of Probiotic Butyricicoccus pullicaecorum Mediates Anticancer Effect on Bladder Urothelial Cells by Regulating Butyrate-Responsive Molecular Signatures"

_diagnostics, 2021, doi:10.3390/diagnostics11122270_

Round 1
Reviewer 1 Report
This paper shows very relevant results and should to my opinion be published.
Herewith some remarks:
- Introduction: "However, gut microbiota are also found in distant organs.....". It is not clear what is meant. For sure not that the gut microbiota itself is found in distant organ, because that's by matter of definition restricted to the gut. So what is meant? The (patho)physiological effects of the gut microbiota with respect to distant organs? Or is it meant that types of microorganisms found in the gut are also found in distant organs? Please explain.
- In the introduction it is not clear why this specific probiotic butyricicoccus pullicaecorum is chosen. Please refer to earlier (pre)clinical results. Also, because the safety of probiotics still remains an issue, it would be helpful to include a reference to the safety of this specific probiotic, e.g. the article "Butyrate Producers as Potential Next-Generation Probiotics: Safety Assessment of the Administration of Butyricicoccus pullicaecorum to Healthy Volunteers" by Boesmans et al.
- Materials and methods: please use superscript and CFU to indicate the amount of probiotics administered: 1*107 CFU in 100 μmL.
- Discussion: Last sentence: "prebiotics" should be replaced by "probiotics"
Author Response
Reviewer #1
We are very grateful to the committee members for affirming this study. Our responses for the comments are:
Comment 1
Introduction: "However, gut microbiota are also found in distant organs.....". It is not clear what is meant. For sure not that the gut microbiota itself is found in distant organ, because that's by matter of definition restricted to the gut. So what is meant? The (patho)physiological effects of the gut microbiota with respect to distant organs? Or is it meant that types of microorganisms found in the gut are also found in distant organs? Please explain.
Response 1
Thanks for the reviewer’s comment. First, we must apologize our carelessness to make an ambiguous sentence. We meant the influences of gut microbiota were not only in the intestinal system, but also in other distant organs, such as lung and breast. Therefore, the dysbiosis in the gut might cause many human diseases, as reported in the references #11 to #14 in the revised version. However, to avoid misunderstanding, we revised the sentence "However, gut microbiota are also found in distant organs....." to "However, the effects of gut microbiota are also found on other organs....."
Comment 2
In the introduction it is not clear why this specific probiotic butyricicoccus pullicaecorum is chosen. Please refer to earlier (pre)clinical results. Also, because the safety of probiotics still remains an issue, it would be helpful to include a reference to the safety of this specific probiotic, e.g. the article "Butyrate Producers as Potential Next-Generation Probiotics: Safety Assessment of the Administration of Butyricicoccus pullicaecorum to Healthy Volunteers" by Boesmans et al.
Response 2
Thanks for the reviewer to give us one reference and we have already added it in the revised edition. According to our previous report (the reference #22, in the revised edition), we thought that the target microbe, “Butyricicoccus pullicaecorum”, would be a gut probiotics. This encouraged us to study its molecular events in other organs. However, in order to clarify our philosophy, we have revised the last paragraph of the section of Introduction to: “In fact, a safe butyrate-producing microbe, Butyricicoccus pullicaecorum, proven by a clinical trial might reduce the cancer progression (21,22)…”.
Comment 3
Materials and methods: please use superscript and CFU to indicate the amount of probiotics administered: 1*107 CFU in 100 μmL.
Response 3
Thanks for the reviewer’s comment. We have revised it.
Comment 4
Discussion: Last sentence: "prebiotics" should be replaced by "probiotics"
Response 4
Thanks for the reviewer’s carefulness. We have revised it.
Reviewer 2 Report
<Summary>
This study investigated the anti-tumor effect of Butyricicoccus pullicaecorum and butylate produced by this probiotics on bladder urothelial cancer. However, there are limitations which should be addressed.
<Comments>
Minor
‘Urothelial bladder cancer’ looks weird. This should be replaced with ‘Urothelial cancer of the bladder’ or ‘bladder urothelial cancer’ throughout the manuscript.
Major
- Page 2, line 66
The authors describe that ‘Supplementation of B. pullicaecorum may provide opportunities for therapeutic intervention in different bladder cancers through secretion of the specific SCFA, butyrate.’ The reviewer is wondering what about supplementation of butyrate itself? Is this harmful? Is this practical? This seems to be easier and more direct than supplementation of B. pullicaecorum.
- Page 2, line 73
The authors describe that ‘All efforts were 73 made to minimize the number of animals and their suffering.’ How the authors statistically determine the number of mice per group. Probably, N=4 is not sufficient to meet statistical power.
- Page 2, line 84
Why the authors selected HT1376 in this study? Please show the rationale of selecting this cell line. The reviewer strongly recommend the authors to use multiple cell lines to obtain consolidated results.
- Page 3, line 130
The authors performed real-time RT-PCR to determine the expression level of three genes. The reviewer is wondering whether these genes are regulated by post-transcriptional modification or not. If yes, RT-PCR should not be performed here.
- Page 5, Figure 5
These pictures do not look mouse bladder. Fixation of bladder was no good. In the fixation process, the authors should distend the bladder, for example, by formalin.
- Page 7, Figure 3E
Western blot does not look good. Re-do.
- Data of real-time RT-PCR
The letters are too small. These are not readable.
- Lack on animal experiments
Please do the experiments to investigate the efficacy of supplementation of Butyricicoccus pullicaecorum in this study.
Author Response
Reviewer #2
We deeply appreciate the reviewer for his insights and questions. Our responses for the comments are:
Minor
‘Urothelial bladder cancer’ looks weird. This should be replaced with ‘Urothelial cancer of the bladder’ or ‘bladder urothelial cancer’ throughout the manuscript.
Response
Thanks for the reviewer’s carefulness. We have corrected it to “urothelial cancer of the bladder”, “bladder urothelial cancer”, or “bladder urothelial cells”, according to the status of the text.
Major
Comment 1 on Page 2, line 66
The authors describe that ‘Supplementation of B. pullicaecorum may provide opportunities for therapeutic intervention in different bladder cancers through secretion of the specific SCFA, butyrate.’ The reviewer is wondering what about supplementation of butyrate itself? Is this harmful? Is this practical? This seems to be easier and more direct than supplementation of B. pullicaecorum.
Response 1
Thanks for the reviewer’s comment. Although oral administration of butyrate has been known to improve mucosa lesion and attenuate the inflammatory profile of intestinal mucosa in colitis in an animal study (J Nutr Biochem. 2012 23(5):430-6), it is not easy to determine the optimal dose of butyrate supplementation, as recently reviewed by Banasiewicz et al. (Prz Gastroenterol. 2020;15(2):119-125). Taken together, reasons from inner layer of intestine, including the rapid absorption of SCFA and the presence of bacterial biofilm and mucus layer, may shorten the effectiveness of oral butyrate administration and difficultly determine the butyrate concentration on the mucous membrane surface. Instead, the administration of probiotics can not only change the gut microbiota, but also stably provide an effective source of butyrate. However, based on the reviewer’s insightful views, we now interest in studying whether the oral administration of butyrate and B. pullicaecorum has a synergistic effect.
Comment 2 on Page 2, line 73
The authors describe that ‘All efforts were made to minimize the number of animals and their suffering.’ How the authors statistically determine the number of mice per group. Probably, N=4 is not sufficient to meet statistical power.
Response 2
In order to ensure the quality and efficiency of the animal experiment, the related animal experiment in this study was supervised by the ''Institutional Animal Care and Use Committee (IACUC)'' in Cathay General Hospital (IACUC No. CGH-IACUC-106-003). It was necessary to explain in detail the design of animal experiment, including the number of animals used in each group. In addition, two major references (Food Chem Toxicol 2014 72:129-37; Sci Rep 2016 6:27572) were referred for the animal numbers. Taken together, we finally controlled the number of animals at n=4 in the reasons of the 3R requirements (replace, reduce, and refine) for animal study and others’ reports.
Comment 3 on Page 2, line 84
Why the authors selected HT1376 in this study? Please show the rationale of selecting this cell line. The reviewer strongly recommend the authors to use multiple cell lines to obtain consolidated results.
Response 3
We appreciated the reviewer’s powerful comment. In fact, we initially cultured two cell lines (HT1376 and BFTC905) for this study and different IC50 values were further detected. But we found that BFTC905 might not be a good cell model for this study because of her derivative source. BFTC905 was propagated from a female Chinese patient (51 years old) with papillary transitional cell carcinoma of the urinary bladder and blackfoot disease. After our detailed discussion, we believed that the blackfoot disease might derivate some confounding and made BFTC905 unsuitable to study the impact of microbiota on bladder cancer. Therefore, we reported the results of HT1376 firstly and we will still try to obtain other available bladder cancer cell lines for more in-depth studies.
Comment 4 on Page 3, line 130
The authors performed real-time RT-PCR to determine the expression level of three genes. The reviewer is wondering whether these genes are regulated by post-transcriptional modification or not. If yes, RT-PCR should not be performed here.
Response 4
Thanks for reviewer’s carefulness. This is something that must be considered. We knew the RNA levels did not necessarily represent the amount of protein. Therefore, we used the immunohistochemical stains to express the levels of most target proteins in mouse bladder tissues (Figure 1A, GPR43; Figure 1B, GPR109B; Figure 1C, FABP4; Figure 3B, BLCAP).
Comment 5 on Page 5, Figure 5
These pictures do not look mouse bladder. Fixation of bladder was no good. In the fixation process, the authors should distend the bladder, for example, by formalin.
Response 5
We thank for the reviewer’s suggestion. Due to the reformatted manuscript by the publisher, we think the Figure 5 issue raised by the reviewer may refer to Figure 1 on page 5, which contains the IHC results of mouse bladder.
The reviewer suggested to distend the bladder by formalin before IHC. As for our data on Figure 1, this was from undistended mouse bladder. After careful literature search, we found that both ways (undistended or distended) for bladder fixation are available. For an example on PNAS (Proc Natl Acad Sci USA. 2019 116(10):4567-74), Figure 1 showed IHC result from undistended bladder. On the other hand, an example on MCP (Mol Cell Proteomics 2015 14(9):2466-78), Figure 3 showed the IHC result from distended bladder samples. Therefore, we think our IHC data of mouse bladder would show the transitional epithelia of undistended bladder. For the reviewer’s reference, we also attached the pictures taken from bladder retrieval from mice.
Comment 6 on Page 7, Figure 3E
Western blot does not look good. Re-do.
Response 6
We appreciated the reviewer’s carefulness. We have produced a clear one to replace the original figure.
Comment 7
Data of real-time RT-PCR
The letters are too small. These are not readable.
Response 7
To improve the quality of all figures and make them readable, we have zoomed in the words appearing on all figures in the revised edition.
Comment 8
Lack on animal experiments
Please do the experiments to investigate the efficacy of supplementation of Butyricicoccus pullicaecorum in this study.
Response 8
Thanks for reviewer’s comment. We have quantified the relative levels of B. pullicaecorum in stools of mice, which haven’t or have been taken B. pullicaecorum administration to demonstrate the efficacy of supplementation of B. pullicaecorum, as shown in supplemental Figure S2 of the revised edition.
Reviewer 3 Report
Wang and colleagues proposed an interesting research article aimed at assessing the anticancer potential of Butyricicoccus pullicaecorum on bladder urothelial cells. For this purpose, the authors performed both animal and in vitro studies to evaluate the effects mediated by butyrate and Butyricicoccus pullicaecorum. Overall, the manuscript is interesting, however, the experimental design is weak as based on a few animal experiments and in vitro evaluation performed in a single cell line. Below are reported some major comments that will improve the manuscript:
1) The in vitro experiments were performed on a single urothelial carcinoma cell line. The results need to be validated in more cell lines or at least in one tumor cell line and a normal one;
2) Have the authors evaluated the effects of supplementation on the drug efficacy? Please address this issue;
3) In the Introduction or Discussion sections, the authors have to better emphasize the importance of microbiota manipulation in cancer. Several studies have demonstrated the benefits of gut microbiota manipulation in different tumors highlighting how microbiota can mediate the effects of targeted and immunotherapies. For this purpose, please see:
- https://doi.org/10.3892/ijo.2021.5255
- https://doi.org/10.1177/1534735419876351
- https://doi.org/10.3892/wasj.2019.13
- https://doi.org/10.1053/j.gastro.2020.11.041
Author Response
1) The in vitro experiments were performed on a single urothelial carcinoma cell line. The results need to be validated in more cell lines or at least in one tumor cell line and a normal one
Response 1
We appreciated the reviewer’s powerful comment. In fact, we urgently purchased another bladder urothelial cell line, 5637 cell line, when we received the comments of reviewer #2. Then, we have added two supplemental figures (Figures S3 and S4) to present the relative expression levels of the butyrate-related genes (GPR43, GPR109B, and FABP4) and the bladder cancer-associated protein (BLCAP) from the butyrate (5 mM)-treated 5637 cells. The results of these gene quantitions have been discussed in the Discussion section (Lines 374-377 and Lines 404-405)
2) Have the authors evaluated the effects of supplementation on the drug efficacy? Please address this issue
Response 2
We really appreciated the reviewer’s powerful comment. This would be helpful for our study. We discuss that B. pullicaecorum supplementation changes the gene expression profile associated with anticancer effect on bladder urothelial cancer. Here, we also found some other reports showed that butyrate could strengthens the anti-cancer effect of cisplatin, which is a standard chemotherapy for advanced bladder cancer (lines 416-419 in the discussion section). Thus, we infer that the synergistic effect of B. pullicaecorum administration can be detected during the period of chemotherapy, even it needs more data to support.
3) In the Introduction or Discussion sections, the authors have to better emphasize the importance of microbiota manipulation in cancer. Several studies have demonstrated the benefits of gut microbiota manipulation in different tumors highlighting how microbiota can mediate the effects of targeted and immunotherapies. For this purpose, please see:
- https://doi.org/10.3892/ijo.2021.5255
- https://doi.org/10.1177/1534735419876351
- https://doi.org/10.3892/wasj.2019.13
- https://doi.org/10.1053/j.gastro.2020.11.041
Response 3
Thanks for the reviewer providing us references. In the Introduction section, we firstly introduce “Some ongoing clinical trials have focused on the signatures of gut microbiota as markers of the efficacy of immune‐checkpoint immunotherapy in bladder cancers from reference #17” (lines 58-59). Then, we discuss the target immunotherapy in the Discussion section (lines 416-423). Briefly, we summarize that “manipulating gut microbiota emerges as a promising adjuvant treatment to enhance the therapeutic effect of cancer immunotherapy”. However, “the therapeutic effect of B. pullicaecorum and its metabolite, butyrate, on the efficacy of anticancer drugs in bladder urothelial cancers warrants future studies”.
Round 2
Reviewer 2 Report
- Page 2, line 84
Why the authors selected HT1376 in this study? Please show the rationale of selecting this cell line. The reviewer strongly recommend the authors to use multiple cell lines to obtain consolidated results.
>The authors did not respond correctly to the reviewer's comment.
Author Response
We appreciated the reviewer’s powerful comment. As our response before, we initially cultured two cell lines (HT1376 and BFTC905) for this study. Unfortunately, BFTC905 might not be a good cell model for this study because of her derivative source.
Now we purchased one new bladder urothelial cell line, 5637 cell line, and further quantified the butyrate-related genes (GPR43, GPR109B, and FABP4) and the bladder cancer-associated protein (BLCAP) from the butyrate (5 mM)-treated 5637 cells. We added supplemental Figures S3 and S4 to discuss the results of these gene quantitations in the Discussion section (Lines 361-364 and Lines 391-392)
Reviewer 3 Report
The authors well-addressed almost all of my previous comments. The experimental design is not complete at all, however, the manuscript could be accepted for publication. Further in vitro experiments are recommended to better evaluate the efficacy of Butyricicoccus pullicaecorum administration in cancer.
Author Response
Thanks for reviewer’s affirmation and we know the shortcomings of our current report. In the 4th edition of revision, we drastically revised the last paragraph (lines 412-425, 4th edition) of the Discussion section and the Conclusions (lines 428-432, 4th edition), based on reviewer’s suggestion. Briefly, we listed two limitations in this study: (1) the possible effect of whole metabolites derived from B. pullicaecorum on bladder cancer was not fully evaluated, especially in combination with other anticancer drugs; (2) the potential interaction of the probiotic B. pullicaecorum with other gut microbiota and their effect on the efficacy of anti-cancer therapy was not addressed.
However, as we stated in the last sentence of Discussion section (lines 424-425), the effects due to B. pullicaecorum supplementation on the bladder urothelial cancers warrant more future studies, even B. pullicaecorum has been a potential next-generation probiotic by 2018 (reference #22, in the 4th edition revision)